# Impacts of Extreme Temperature and Precipitation on Crops during the Growing Season in South Asia

Xinyi Fan [1,2], Duoping Zhu [2,3], Xiaofang Sun [1], Junbang Wang [2,*], Meng Wang [1], Shaoqiang Wang [2] and Alan E. Watson [4]

1   School of Geography and Tourism, Qufu Normal University, Rizhao 276800, China
2   National Ecosystem Science Data Center, Key Laboratory of Ecosystem Network Observation and Modeling, Institute of Geographic Sciences and Natural Resources Research, Chinese Academy of Sciences, Beijing 100101, China
3   Institute of Desertification Studies, Chinese Academy of Forestry, Beijing 100091, China
4   USDA Forest Service, Rocky Mountain Research Station, Missoula, MT 59801, USA
*   Correspondence: jbwang@igsnrr.ac.cn; Tel.: +86-(10)-6487-4229

**Abstract:** South Asia, one of the most important food producing regions in the world, is facing a significant threat to food grain production under the influence of extreme high temperatures. Furthermore, the probability of simultaneous trends in extreme precipitation patterns and extreme heat conditions, which can have compounding effects on crops, is a likelihood in South Asia. In this study, we found complex relationships between extreme heat and precipitation patterns, as well as compound effects on major crops (rice and wheat) in South Asia. We also employed event coincidence analysis (ECA) to quantify the likelihood of simultaneous temperature and crop extremes. We used the Enhanced Vegetation Index (EVI) as the primary data to evaluate the distinct responses of major crops to weather extremes. Our results suggest that while the probability of simultaneous extreme events is small, most regions of South Asia (more than half) have experienced extreme events. The regulatory effect of precipitation on heat stress is very unevenly distributed in South Asia. The harm caused by a wet year at high temperature is far greater than that during a dry year, although the probability of a dry year is greater than that of a wet year. For the growing seasons, the highest significant event coincidence rates at a low EVI were found for both high- and low-temperature extremes. The regions that responded positively to EVI at extreme temperatures were mainly concentrated in irrigated farmland, and the regions that responded negatively to EVI at extreme temperatures were mostly in the mountains and other high-altitude regions. Implications can guide crop adaptation interventions in response to these climate influences.

**Keywords:** extreme temperature; extreme precipitation; crop; South Asia

## 1. Introduction

The impacts of climate change and extreme weather events on agricultural productivity observed over the last two decades are projected to continue into the future [1–3]. Global warming is leading to an increased rate of evapotranspiration, increasing intensity and duration of drought [4], and more frequent and extended extreme weather events [5–7], all of which can affect crop yields. Temperature and precipitation are closely related to crop growth and crop yield. Therefore, better understanding the effects of temperature and precipitation extremes is important for the agricultural sector for mitigation and adaptation to climate change, and to increase resilience and resistance to negative impacts on food production.

In South Asia, average annual temperatures in Bangladesh, India and Myanmar have continued to rise, annual precipitation has increased and high temperature heat waves, extreme precipitation events, extreme drought and storm surge events all showed increasing trends in most regions, potentially leading to food insecurity in the region [8–13].

Extreme temperature and precipitation events, and their threat to food security, are spatially heterogeneous, but becoming more serious in some areas of South Asia, especially in Bangladesh, India and Myanmar [14,15]. For example, the effects of temperature variability and extremes on crops showed an intra-regional difference in India [16]. In Bangladesh, location-specific climatic hazards were identified to be rainfall, temperature and combined effects [17]. Myanmar also showed uneven effects of dry-wet spatial variation on crops [18]. The frequency of extreme weather events is projected to increase as climate warming intensifies [19]. With the prospect of increasing frequencies and/or intensities of extreme meteorological events in South Asia [20–23], farmers face serious challenges to adapt to and mitigate climate change [24]. Under the circumstances, it is imperative to investigate the effects of temperature and precipitation extremes and their combined effects on crop yield.

Generally speaking, the occurrence of an extreme weather event, such as extreme temperature or extreme precipitation patterns, may cause severe crop damage, but when extreme temperature and precipitation conditions are intertwined, the damage caused by extreme weather may be moderated [25–27]. For example, during high temperatures, plants can emit water through high evapotranspiration to prevent heat damage [28,29]. Up until now, few studies have investigated the combined effect of extreme temperature and extreme precipitation patterns on crop yield over a global scale [30–32].

Remote sensing has been widely used to quantify effects of climate change on vegetation on local to global scales. The Normalized Difference Vegetation Index (NDVI) from Global Inventory Modeling and Mapping Studies (GIMMS) has been shown to be a robust and reliable measure to represent real responses of vegetation to climate variability [33,34], and can be used to quantify effects from extreme climates due to its long time series spanning from 1981 to 2015 [35–38]. The Enhanced Vegetation Index (EVI) was developed to minimize atmospheric- and soil-background effects [39] and be more sensitive to high biomass areas, all of which pertain to vegetation in tropical regions [40–42]. The EVI product from MODIS (Moderate resolution Imaging Spectroradiometer) has been improved in data quality by removing the major sensor degradation impacts [43] and implementing several improvements in its retrieval algorithm [44]. Therefore, both EVI and NDVI provide a chance to analyze extreme temperature and precipitation effects on crop growth, quantified by the Vegetation Index (VI), maximizing their respective strengths.

This study aims to (i) quantify the probability and likely extent of extreme temperature events under different precipitation amounts and their combined impact on crops, and (ii) examine the impacts of extreme temperature events on crop growth through event coincidence analysis. This study provides a framework to quantify the impact of compound weather extremes on crops, which can be useful for prioritizing needs for mitigation and adaptation through cropping systems management.

## 2. Materials and Methods

### 2.1. Study Area

The study area includes three low-latitude countries: Bangladesh, India and Myanmar in South Asia. This region has a tropical monsoon climate with mean annual temperature of 27 °C, an annual precipitation between 1500 and 2000 mm and shows both an obvious dry and rainy season.

In the context of frequent climate extremes, recurrent and consecutive droughts have led to uncertainty about rainfed agriculture and its sustainability in India and Myanmar [45–47]. Meanwhile, Bangladesh is a flood-prone country and often experiences devastating floods during the monsoon season that cause damage to crops and property.

About 65% of the study area is cropland, with 8.84%, 87.00% and 4.16% of the entire study area in Bangladesh, India and Myanmar, respectively (Figure 1). The total amount of arable land in India ranks first in Asia, and India is one of the world's largest food producers. The Ganges Plain and Deccan Plateau are flat and fertile, and they are the main farming areas of India. The terrain of Myanmar is high in the north and low in the south. The central river valley and plains are very suitable for farming.

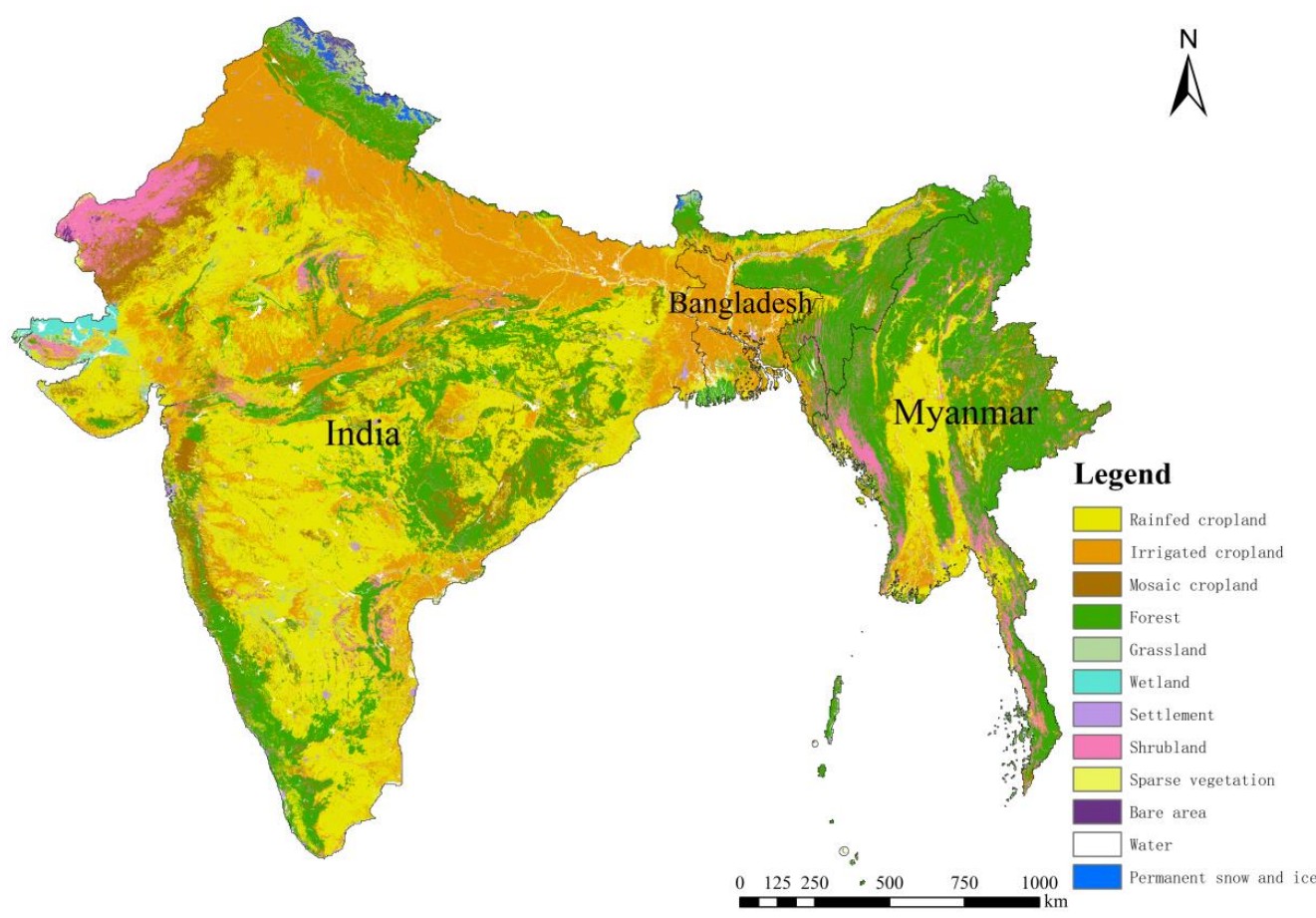

**Figure 1.** Types of land use in the study area.

The rice cropping systems in India are generally divided into three types: Aus, Kharif and Rabi, related to the strong influence of the seasonal pattern of precipitation [48]. Aus is the summer crop, starting in March and going to June. Kharif is the rainy season crop starting in June and going to November, and Rabi is the winter crop, or dry season crop, starting in December and extending to March. In Myanmar, there are two distinct cropping seasons [49]: Rabi (mid-November to mid-March) and Aus (mid-July to mid-November). In a calendar year, Bangladesh has three main crop growing seasons, that is, the monsoon period between June and October, the winter period between November and February and the summer period between March and June [50]. In Bangladesh, India and Myanmar, wheat is generally sown in November-December and harvested in March-April.

Agriculture in the study area, therefore, is mainly the rice-wheat double or triple cropping system, varying somewhat across the three countries (Table 1). Considering India has the largest area of cropland, which is ten times that of Myanmar, and twenty times that of Bangladesh (Figure 1), the cropping seasons of India were applied to the other two countries and the same periods were applied to avoid the impacts of the differences in growing periods. The final cropping seasons for this study include Rabi (January–February), Aus (May–June), and Kharif (August–September).

**Table 1.** The crop calendar for the study region of Bangladesh, India and Myanmar. The blue color is for crop sowing, green for crop growing and yellow for crop harvest.

| Country | Crop | Variety | Jan. | Feb. | Mar. | Apr. | May | Jun. | Jul. | Aug. | Sep. | Oct. | Nov. | Dec. |
|---|---|---|---|---|---|---|---|---|---|---|---|---|---|---|
| Bangladesh | rice | Aus | | | sowing | sowing | growing | growing | harvest | harvest | | | | |
| | | kharif | harvest | | | | | | sowing | sowing | | growing | growing | harvest |
| | | Rabi | sowing | growing | growing | | harvest | harvest | | | | | | sowing |
| | wheat | | growing | growing | harvest | harvest | | | | | | | sowing | sowing |
| India | rice | Aus | | | sowing | sowing | growing | growing | harvest | harvest | | | | |
| | | Kharif | | | | | | sowing | sowing | growing | growing | harvest | harvest | |
| | | Rabi | growing | growing | harvest | harvest | harvest | | | | | | sowing | sowing |
| | wheat | | growing | growing | harvest | harvest | | | | | | | sowing | sowing |
| Myanmar | rice | Aus | harvest | | | | | | sowing | sowing | growing | growing | harvest | |
| | | Rabi | growing | growing | harvest | harvest | | | | | | | sowing | sowing |
| | wheat | | growing | growing | harvest | harvest | | | | | | | sowing | sowing |

### 2.2. Data

The datasets used in this study include EVI and meteorological data and are summarized in Table 2. The EVI data were applied to quantify the impact of extreme weather on crop growth and is derived from the Global Vegetation Indices Monthly L3 data product of MODIS (MOD13C2) [51]. It has a spatial resolution of 0.05° latitude and longitude (near 5 km) from 2000 to 2018. The dataset was resampled to a spatial resolution of 5 km with an equal area map projection and monthly temporal resolution [52]. For this paper, the results are primarily determined from examining EVI-based analyses. The GIMMS-based NDVI were also analyzed, taking advantage of the long-term observations spanning from 1981 to 2015, and improved data quality by accounting for biases such as calibration loss, orbital drift, volcanic eruptions [53,54], etc. The results from NDVI are shown in Supplementary Figures S1 and S2.

**Table 2.** Datasets used in this study.

| | Product | Range | Temporal Resolution | Spatial Resolution | Resampling |
|---|---|---|---|---|---|
| NDVI | GIMMS NDVI3g | 1982–2015 | Monthly | 5000 m | |
| EVI | MOD13C2 | 2000–2018 | Monthly | 5000 m | Monthly |
| Temperature and Precipitation | ERA5 | 1982–2018 | 8-days | 0.25° | 5000 m × 5000 m |
| Land use | ESA CCI | 2020 | Year | 300 m | |

ERA5 [55] is the fifth generation atmospheric reanalysis of the global climate and is produced by the Copernicus Climate Change Service (C3S) at the European Centre for Medium-Range Weather Forecasts (ECMWF) [56]. The temperature and precipitation data were derived from the ERA5, including the hourly air temperature at 2 m height and total precipitation with a spatial resolution of 0.25° latitude and 0.25° longitude from 1979 to the present. The data were resampled to 5 km spatial resolution and calculated at a monthly temporal resolution to match with EVI data.

Land cover data are obtained from the European Space Agency (ESA) and the product is an annual ESA CCI (Climate Change Initiative) land cover map of the world at 300 m resolution [57], which has undergone preprocessing such as atmospheric calibration and geometric correction [58]. Here, the data of the most recent available year (2020) are used.

### 2.3. Methods

#### 2.3.1. Frequency and Extent of Climate Extreme Events

Monthly precipitation anomaly percentage ($P$) was applied to portray precipitation year conditions [59].

$$P = \frac{P_m - \overline{P}}{\overline{P}} \cdot 100\% \tag{1}$$

where $P_m$ is the monthly total precipitation, and $\overline{P}$ is the monthly precipitation averaged from 1982 to 2018. When $P$ is greater than 50%, it is a wet year; when $P$ is less than $-50\%$, it is a dry year, otherwise, it is a normal precipitation year.

The 10th and 90th percentiles were applied to indicate extreme temperature events [60]. The concurrence frequency and extent of extreme temperature and precipitation were measured by the composition of the monthly total precipitation anomaly ($P$) and the monthly mean temperature ($T$), and nine conditions were indicated, that is:

HTHP: $T > $ 90th and $P > 50\%$,
HTNP: $T > $ 90th and $-50\% \leq P \leq 50\%$,
HTLP: $T > $ 90th and $P < -50\%$,
NTHP: 10th $\leq T \leq$ 90th and $P > 50\%$,
NTNP: 10th $\leq T \leq$ 90th and $-50\% \leq P \leq 50\%$,
NTLP: 10th $\leq T \leq$ 90th and $P < -50\%$,
LTHP: $T < $ 10th and $P > 50\%$,
LTNP: $T < $ 10th and $-50\% \leq P \leq 50\%$,
LTLP: $T < $ 10th and $P < -50\%$.

The frequency and coverage area were calculated for the above nine weather conditions through the monthly data for each period from 1982 to 2015.

### 2.3.2. Measuring Influence of Extreme Climate Events on Crop Growth

Influence was measured through the deviation of EVI due to extreme climate events by comparing with the corresponding EVI under the normal $T$ and $P$. The measurements include regional mean, standard deviation and degree of deviation.

### 2.3.3. Event Coincidence Rate between Extreme Temperature and Extreme EVI

The event coincidence analysis (ECA) computes the event coincidence rate (ECR), that is, the empirical fraction of simultaneous events in two series [61,62]. By definition, ECR has a value of between 0 and 1, where the closer to 0, the less likely the events in both series are to occur at the same time (indicating that there is no corresponding instantaneous statistical relationship), and the closer the ECR is to 1, the more likely the two events in a series will always happen at the same time. The statistical significance of the ECRs was obtained using a simple analytical significance test against the null hypothesis of two independent Poisson processes with low event rates, using a significance level of $\alpha = 0.05$ [61].

In this study, the ECA was applied to measure consistency between the two extreme time series of temperature and EVI, which was defined by the 10th and 90th percentiles of the time series and were applied to define the extremely low and extremely high EVI for each growing period [60]. To distinguish between the presence or absence of an extreme event at the given temporal aggregation level, the positive temperature event was used to denote extreme high temperature and negative temperature event to represent extreme low temperature. The strength of defining an anomaly that simultaneously exceeds a univariate threshold as a joint anomaly is that it provides a spatially consistent and transparent measure without masking the true joint occurrence of EVI and temperature. Taking different types of events in both time series (EVI and temperature) leaves us with four possible event combinations to be considered for three growing seasons:

(i)    Both 2 m temperature and EVI are greater than their respective empirical 90% quantiles (in the following referred to as $T_{90}$–$V_{90}$)
(ii)   Both 2 m temperature and EVI are lower than their 10% quantile ($T_{10}$–$V_{10}$)
(iii)  2 m temperature is lower than its 10% and EVI is greater than its 90% quantile ($T_{10}$–$V_{90}$)
(iv)   2 m temperature is greater than its 90% and EVI is lower than its 10% quantile ($T_{90}$–$V_{10}$)

For the specific formula on the ECA, please refer to Appendix A [61]. Python was applied to code an algorithm program to perform ECA and calculated the SCR for the two extreme time series of temperature and vegetation index.

The above data and method can be summarized in the framework in Figure 2.

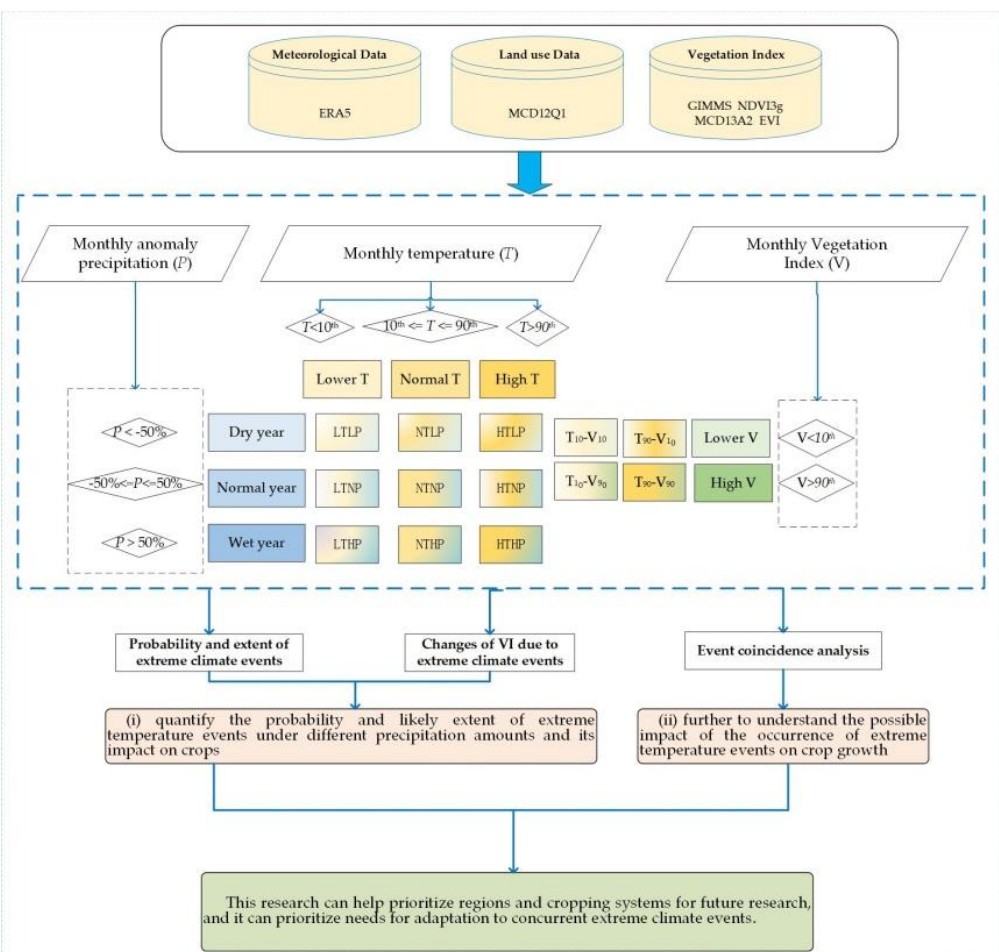

**Figure 2.** The framework to analyze the impacts of extreme temperature and precipitation on crops in Bangladesh, India and Myanmar.

## 3. Results

### 3.1. Probabilities of Extreme Climate Events

The nine extreme conditions defined in Section 2.3.2 occur in varying probability among the three growing seasons for the study area (Table 3 and Figure 3). In the three growing seasons, the probability of extreme high and low temperature is 10.29% and the probability of occurrence of dry years is much greater than that of wet years. The probability of dry years from January to February is as high as 52.73%, which is twice that of wet years or even normal precipitation years. Among the combinations without normal temperature or normal precipitation, in other words, among the four extreme weather combinations (HTLP, HTHP, LTLP, LTHP), the combination of higher temperature in a dry year has the highest probability, reaching 6.81%, 7.19%, and 4.19%, in the three growing seasons, respectively.

**Table 3.** During the 1982–2015 growing seasons, the probability of occurrence of various extreme meteorological events in the study area and the proportion of areas that experienced various extreme meteorological events.

|  | Probability (%) | | | Distribution Area (%) | | |
|---|---|---|---|---|---|---|
|  | **Jan–Feb** | **May–Jun** | **Aug–Sep** | **Jan–Feb** | **May–Jun** | **Aug–Sep** |
| LTLP | 4.56 | 0.44 | 0.61 | 92.14 | 13.02 | 14.50 |
| LTNP | 2.18 | 4.02 | 6.16 | 62.52 | 84.95 | 91.64 |
| LTHP | 3.56 | 5.84 | 3.52 | 84.63 | 86.78 | 93.38 |
| NTLP | 41.35 | 25.79 | 11.77 | 100.00 | 97.57 | 96.60 |
| NTNP | 19.23 | 37.87 | 57.66 | 100.00 | 100.00 | 100.00 |
| NTHP | 18.82 | 15.76 | 9.98 | 100.00 | 99.61 | 93.20 |
| HTLP | 6.81 | 7.19 | 4.19 | 99.01 | 97.11 | 85.52 |
| HTNP | 1.68 | 2.54 | 5.97 | 55.47 | 86.73 | 93.86 |
| HTHP | 1.80 | 0.57 | 0.13 | 67.16 | 25.75 | 5.41 |

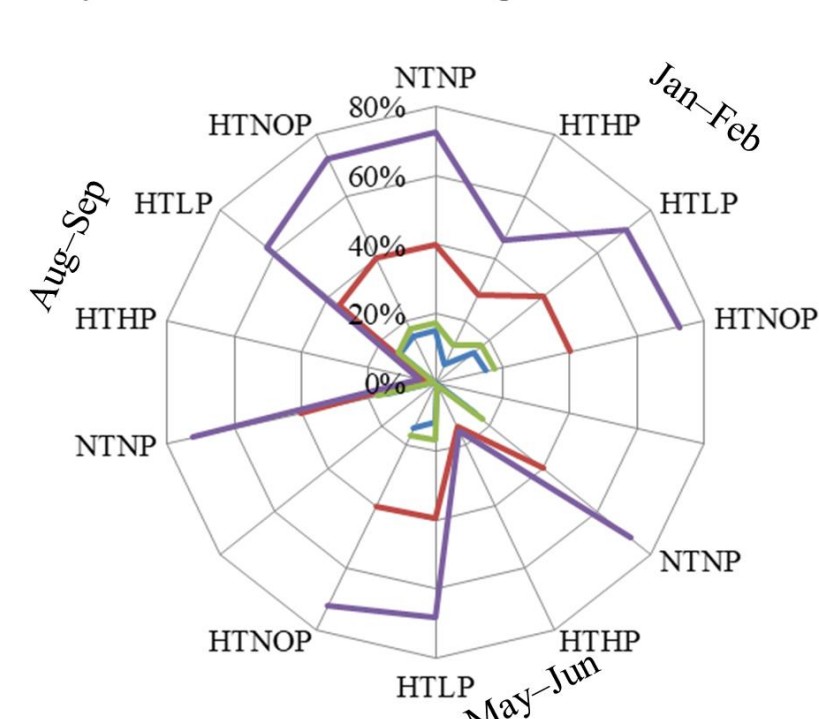

**Figure 3.** The cropland percentage of area that experienced extreme high temperature events during growing seasons in Bangladesh, India, Myanmar and the whole area from 2000 to 2018. (Abbreviations: NTNP–Normal temperature and normal precipitation, HTLP–Higher temperature and lower precipitation, HTNOP–Higher temperature without precipitation, HTHP–Higher temperature and higher precipitation).

Although the probability of extreme weather events is far less than normal weather, more than half of the cropland in the study area has experienced such extreme weather. The highest amount of cropland exposed to extreme temperatures from Jan to Feb was 73.88% and 79.76% for extreme high temperatures and low temperatures, respectively. Only 61.59% (66.51%) and 69.86% (61.60%) of cropland was exposed to extreme high temperatures and extreme low temperatures from May to June (August to September). The largest area exposed to extreme precipitation was also in January–February, reaching 83.93% and 97.05% in wet and dry years, respectively. In the other two growing seasons, the area exposed to wet and dry years was between 64.00% and 70.72%.

We further explored the percentage of crop areas that suffered from extreme high temperature (Figure 3). From the perspective of the growing seasons, the extreme high temperature is most frequent in the Jan–Feb season, in the dry years (70.94%), while the area affected is only 45.85% in wet years. India was exposed to more extreme high temperature events with an area percentage of 36.16%, followed by Bangladesh and Myanmar.

### 3.2. Changes of EVI Due to Extreme Climate Events

The changes of EVI due to extreme climate events were analyzed by comparing to those at normal temperature conditions (NTNP) (Figures 4 and 5) and we found that the high temperatures significantly affected crop growth, according to one-way ANOVA (Table 4). Specifically, under the condition of extreme high temperatures without considering differences in precipitation (HTNOP), the EVI would decrease 1.85%, 8.25% and 2.36%, respectively, in the three growing seasons of Jan–Feb, May–Jun and Aug–Sep. If considering the condition of higher precipitation (HTHP), the negative effect would be 3.85%, 32.29% and 24.19%, while under the condition of HTLP, crop growth would decrease 4.26%, 15.15% and 6.52%, respectively, in three seasons according to EVI. The results revealed that the effects of high temperature extremes occur mainly in May–Jun and are usually four times and two times greater for crop growth under the combination of flooding or drought, and to a larger extent for solely high temperatures.

### 3.3. Event Coincidence Analysis

The occurrence of events coincidence with the different SCRs are presented in Figure 6. The event coincidence occurrence area of extreme temperature and EVI showed a very extensive and wide distribution in the study region (Figure 7). As shown in Table 5, during January–February, the average probability of the $T_{90}$–$V_{90}$ is 14.52%, which is three times the probability of $T_{90}$–$V_{10}$ (Figure 6a). However, the SCR area of the $T_{90}$–$V_{90}$ combination is smaller than that of the $T_{90}$–$V_{10}$ combination, accounting for 42.50% and 57.50% of the study area, respectively. The significant SCRs (SCR > 0.5) of $T_{90}$–$V_{90}$ were found in the Deccan Plateau of India and central Myanmar, while the significant SCRs of $T_{90}$–$V_{10}$ are mainly distributed along the Siwalik Hills, northern Bangladesh and southern Myanmar.

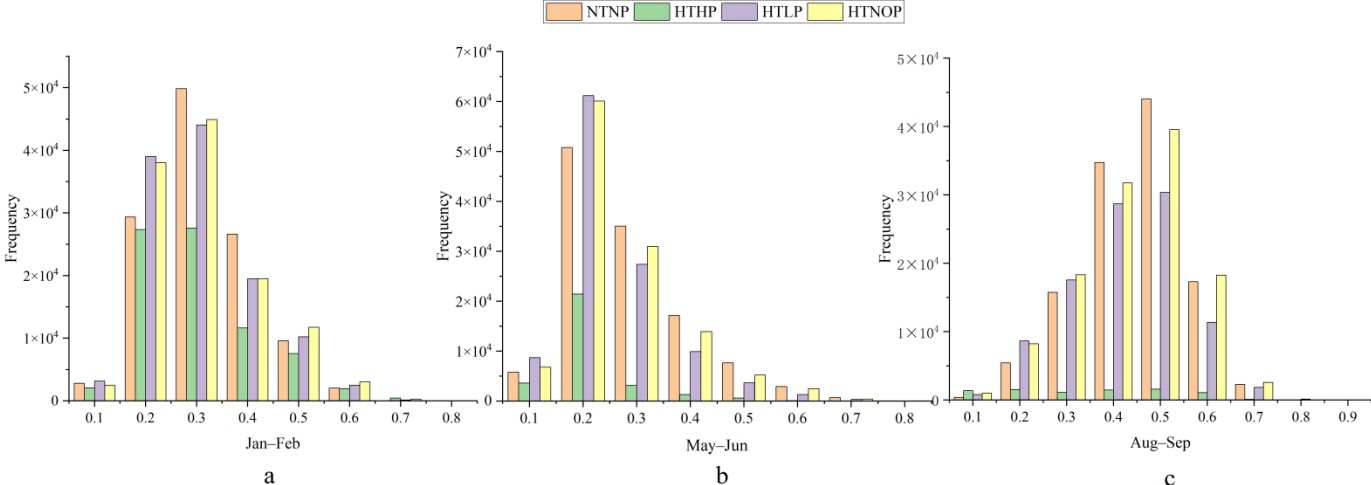

**Figure 4.** Frequency distribution of EVI under high temperature extreme events combined higher, normal and lower precipitation in three growing seasons of Jan–Feb (**a**), May–Jun (**b**) and Aug–Sep (**c**). (Abbreviations: NTNP –Normal temperature and normal precipitation, HTLP–Higher temperature and lower precipitation, HTNOP–Higher temperature without precipitation, HTHP-Higher temperature and higher precipitation).

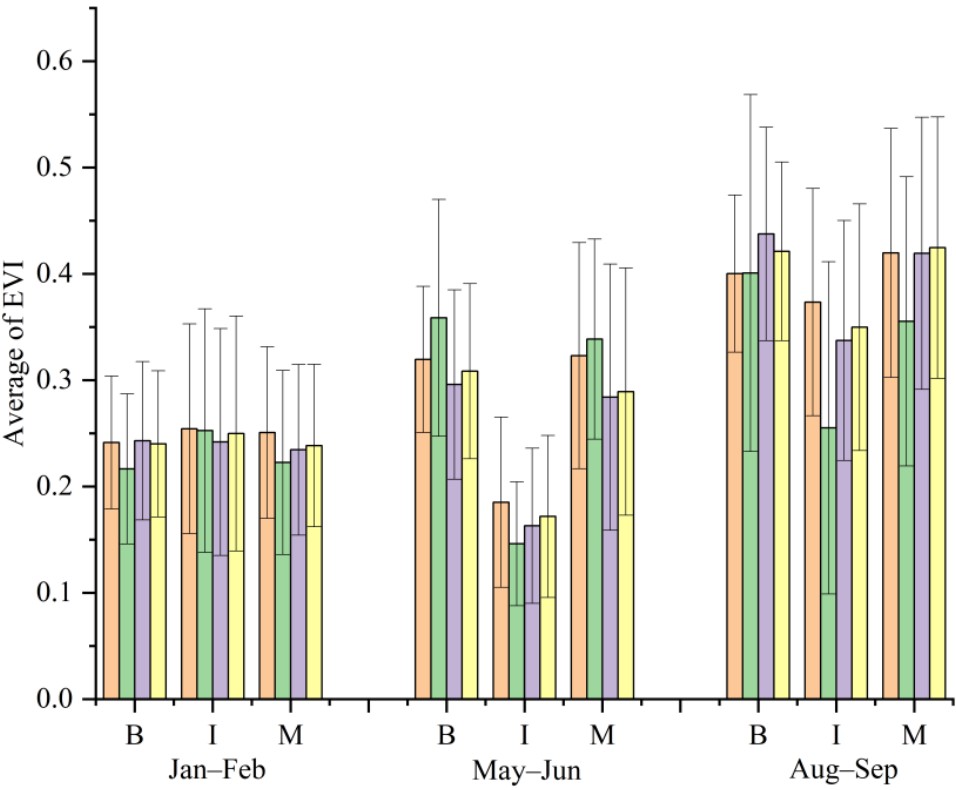

**Figure 5.** The comparison of the averaged EVI for different meteorological combination events in Bangladesh (B), India (I) and Myanmar (M), with error bars representing standard deviations. (Abbreviations: NTNP–Normal temperature and normal precipitation, HTLP–Higher temperature and lower precipitation, HTNOP–Higher temperature without precipitation, HTHP–Higher temperature and higher precipitation).

**Table 4.** The Analysis of variance (ANOVA) of the impacts of extreme high temperatures on EVI during three growing seasons.

| 2000–2018 | Grop | Sum | Mean | Variance | | SS | df | MF | F | α | F Crit |
|---|---|---|---|---|---|---|---|---|---|---|---|
| January–February | NTNP | 31,873.58 | 0.26 | 0.01 | Within groups | 9.16 | 3.00 | 3.05 | 292.58 | 0.00 | 2.60 |
| | HTHP | 20,033.14 | 0.25 | 0.01 | | | | | | | |
| | HTLP | 30,092.49 | 0.25 | 0.01 | Between groups | 4566.31 | 437,428.00 | 0.01 | | | |
| | HTNOP | 31,191.08 | 0.26 | 0.01 | | | | | | | |
| May–June | NTNP | 28,007.44 | 0.23 | 0.01 | Within groups | 165.15 | 3.00 | 55.05 | 5211.33 | 0.00 | 2.60 |
| | HTHP | 4777.05 | 0.16 | 0.01 | | | | | | | |
| | HTLP | 22,283.32 | 0.20 | 0.01 | Between groups | 4040.82 | 382,534.00 | 0.01 | | | |
| | HTNOP | 25,670.35 | 0.21 | 0.01 | | | | | | | |
| August–September | NTNP | 47,735.72 | 0.40 | 0.01 | Within groups | 95.39 | 3.00 | 31.80 | 2377.91 | 0.00 | 2.60 |
| | HTHP | 2472.22 | 0.30 | 0.03 | | | | | | | |
| | HTLP | 36,986.52 | 0.37 | 0.01 | Between groups | 4640.95 | 347,078.00 | 0.01 | | | |
| | HTNOP | 46,522.06 | 0.39 | 0.01 | | | | | | | |

Abbreviations: NTNP–Normal temperature and normal precipitation, HTLP–Higher temperature and lower precipitation, HTNOP–Higher temperature without precipitation, HTHP–Higher temperature and higher precipitation.

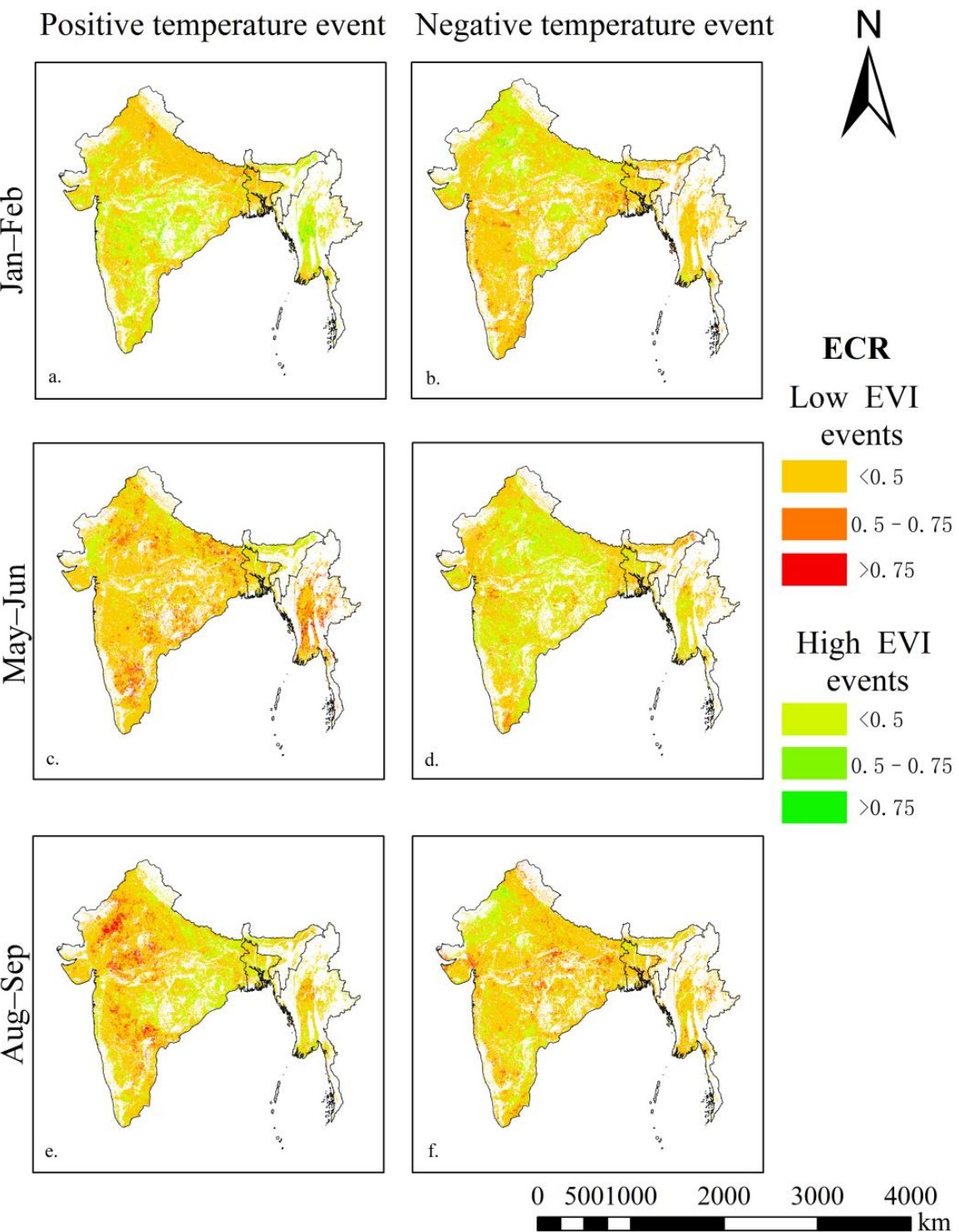

**Figure 6.** Spatial patterns of significant event coincidence rates (SCRs) between different combinations of low and high temperature and EVI extremes during the three considered time periods ((**a**,**c**,**e**) and (**b**,**d**,**f**) represent extreme EVI under extreme high and extreme low temperatures in Jan-Feb, May-Jun and Aug-Sep, respectively, with red representing extreme low EVI values and green representing extreme high EVI values).

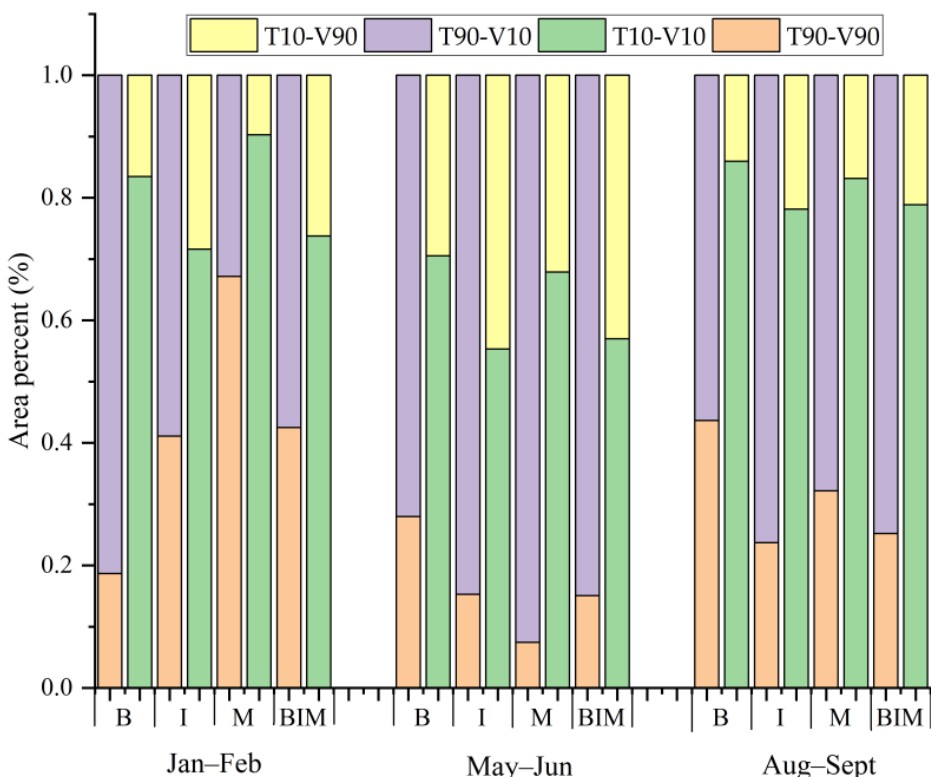

**Figure 7.** The area proportions of significant event coincidence rates (SCRs) between different combinations of extremes; lower (the 10th quantile) and higher (the 90th quantile) temperature ($T_{10}$ and $T_{90}$) and vegetation index ($V_{10}$ and $V_{90}$) in the three growing periods for Bangladesh (B), India (I), Myanmar (M) and the entire region (BIM).

**Table 5.** Significant event coincidence rates (SCRs) between different combinations of extremes; lower (the 10th quantile) and higher (the 90th quantile) temperature (T10 and T90) and vegetation index (V10 and V90) in the three growing periods for Bangladesh (B), India (I), Myanmar (M) and the entire region (BIM).

| Significant Event Coincidence Rates (%) | Jan–Feb | | | | May–Jun | | | | Aug–Sep | | | |
|---|---|---|---|---|---|---|---|---|---|---|---|---|
| | **B** | **I** | **M** | **BIM** | **B** | **I** | **M** | **BIM** | **B** | **I** | **M** | **BIM** |
| $T_{90}$–$V_{90}$ | 5.10 | 13.80 | 27.00 | 14.52 | 8.76 | 4.60 | 2.89 | 4.57 | 14.49 | 7.39 | 10.08 | 7.83 |
| $T_{90}$–$V_{10}$ | 13.44 | 4.65 | 2.03 | 4.78 | 6.94 | 15.56 | 30.30 | 16.58 | 5.04 | 14.07 | 8.59 | 13.29 |
| $T_{10}$–$V_{10}$ | 11.29 | 8.13 | 10.66 | 8.48 | 8.36 | 7.20 | 7.03 | 7.26 | 13.74 | 13.24 | 15.32 | 13.50 |
| $T_{10}$–$V_{90}$ | 4.41 | 8.94 | 2.81 | 8.14 | 8.30 | 14.93 | 10.00 | 14.12 | 4.01 | 6.79 | 5.10 | 6.47 |

The average probability of the $T_{10}$–$V_{10}$ is similar to that of the $T_{10}$–$V_{90}$ (Figure 6b). The difference is that the SCRs of the $T_{10}$–$V_{10}$ combination accounted for 73.74% of the study area, which was three times that of the $T_{10}$–$V_{90}$. The SCRs of $T_{10}$–$V_{10}$ are mainly concentrated in the Ganges Plain, Indus Plain and Central Mountain Plateau, of $T_{10}$–$V_{10}$ are widely distributed in the study area, where the significant SCRs are located in the high mountains such as the Kasha Mountains, Naga Hills, Western Ghats and Eastern Ghats.

Unlike the Jan–Feb season, the average probability of $T_{90}$–$V_{10}$ in May–Jun was 16.58%, and was three times higher than that of $T_{90}$–$V_{90}$ (Figure 6c). Here, the SCRs cover 84.93% of the total area and concentrate in five distinct regions: The Deccan Plateau, Eastern Ghats, along the Ganges Plain, the Indian Desert and the Bago Mountains in Myanmar, while the SCRs for $T_{90}$–$V_{90}$ are concentrated only along the lower Indus, lower Ganges and Brahmaputra rivers.

The probability of the T10–V90 (14.12%) is twice that of T10–V10 in May–Jun (Figure 6d). The area of SCRs accounted 43.01% of the whole study area in May–Jun, which is the maximum area appearing (or happening) among the three seasons. Notably, these distributions were somewhat similar to the T90–V90 observations between Jan and Feb: SCRs were concentrated in central India and Myanmar. Although the distribution of T10–V10 is relatively loose, its SCRs are concentrated in southern India and the Sahyadri Mountains.

In Aug–Sep, the extreme values of EVI at extreme temperatures also showed the opposite in space. In extreme high temperature, an extremely high EVI is more widely distributed in the east of India, and an extremely low EVI is distributed in the west, while in extreme low temperature, an extremely high EVI is concentrated in western India, and an extremely low EVI is distributed in the west of India. In particular, the SCRs of $T_{90}$–$V_{10}$ accounted for 74.81% of the study area, and among them, the most significant SCRs can be found on the western coast of India. The SCRs of $T_{90}$–$V_{90}$ are mainly distributed between the Ganges River and the Godavari River (Figure 6e). The significant SCRs of $T_{10}$–$V_{10}$ are scattered along the Ganges River (Figure 6f), the SCRs of $T_{10}$–$V_{90}$ are concentrated in the Indus Plain (Figure 6f) and a small agglomeration is concentrated in the Eastern Ghats, accounting for 21.13% of the study area as a whole.

It could be found that the both $T_{90}$–$V_{10}$ and $T_{10}$–$V_{10}$ showed the highest probability and widest area according to their SCRs. The spatial distribution of SCRs varied with the regions. The regions where the EVI positively responded to extreme temperature were mainly in irrigated farmland, while the regions where the EVI responded negatively to extreme temperature were mostly in the mountains and other high-altitude regions.

## 4. Discussion

### 4.1. Extreme Climate Events under Global Warming

The frequency and intensity of extreme high temperatures has undoubtedly increased in the study area, as reported by IPCC AR6 [63]. At the same time, heat extremes often were accompanied by increasing extreme precipitation patterns such as drought or flooding, and the combined extreme events occur with a higher probability [64,65]. In addition, crops in the study area were extensively exposed to extreme weather during the growing season as shown in Table 3 and Figure 3. These extreme weather conditions have resulted in widespread damage to crop growth and yield (Figures 4 and 6), and threatened crop safety, not only for people in this study area, but also for neighboring countries. However, this study is the first to quantify probability of the combined effect and spatial distribution for this region.

Extreme heat under global warming may associate with insufficient precipitation, which is generally the main factor controlling drought onset, resulting in insufficient soil moisture, episodic combination of soil moisture supply deficit and atmospheric vapor demand requirements that exert water stress on the crop growth [66]. On the other hand, continued global warming is projected to further intensify the global water cycle, including extremely high precipitation, droughts and compound extremes [67]. Most South Asian countries have experienced extreme precipitation, with more spatial heterogeneity within sub-regions, such as in India [68–71].

### 4.2. The Mechanism of Crop Yield Reduction Caused by Extreme Climate

When extreme heat and moisture conditions are intertwined, the combined hazard to crops is far greater than the single hazard of extreme heat (Figure 4). Although the probability of extreme drought is greater than that of extreme high precipitation (Table 1), the damage intensity and area of crops in rainy years are more serious than those in dry years, and the average vegetation index further supports this conclusion (Figure 5). Severe water stress can lead to crops slowing net photosynthesis and shortening the growth period, in particular when droughts persist for an extended period or occur during key plant developmental stages, eventually leading to crop failure [72,73]. On the contrary, during periods of high precipitation, increasing temperatures not only reduce the efficiency

of photosynthesis, but also result in a faster development of crops; this leads to a shorter life cycle, resulting in smaller crops, then a shorter grain filling period, and finally, to lower yields [74]. Once flood stress occurs, the limited function of root systems leads to weakened respiration and even root damage, while nutrient loss and soil erosion reduce nutrient supply, thus affecting crop growth.

Under the trend of global warming, the possibility of extreme high temperature in South Asia is much greater than that of extreme low temperature as shown in Table 3. It has been found that in South Asia, low productivity mostly was caused by low temperatures, resulting from the enzymatic reaction of plants, not by abnormal low temperature-induced freezing injury [75–77]. Therefore, this study mainly considered the extreme high temperature that generally is defined as temperature above a critical threshold for a specific crop [78], but here the pixel-scale temperature at the 90th quantile was used. On the other hand, duration of heat exposure is another aspect that results in productivity losses by varying mechanisms such as causing thermal denaturation of proteins, damaging tissues and inducing drought conditions [79–84]. Therefore, the duration of heat extreme events and their effects on crop growth should be further studied in the future.

### 4.3. Similarities and Differences in Vegetation Index

Remote sensing-based analysis is a common method to illustrate the effects of extreme climate events on actual crop growth for rainfed or irrigated cropland. We applied the EVI to quantify the combined effects of extreme temperature and precipitation on crops, and used the NDVI as a comparison. The results showed that the EVI is more sensitive to quantifying the effects of extreme climate events than the NDVI (Supplementary Figure S1). Although the average deviation of the vegetation index at different precipitation levels under high temperature were both mainly negative deviations, the extent of negative deviation from the EVI was more intense and showed a more clear spatial pattern than that from the NDVI (Figure 8 and Supplementary Figure S2). The largest negative deviation of EVI was −19.83% in the wet years in Aug–Sep (Figure 9), while the largest negative deviation from the NDVI was only −3.02% in the wet years in May–Jun (Supplementary Figure S3).

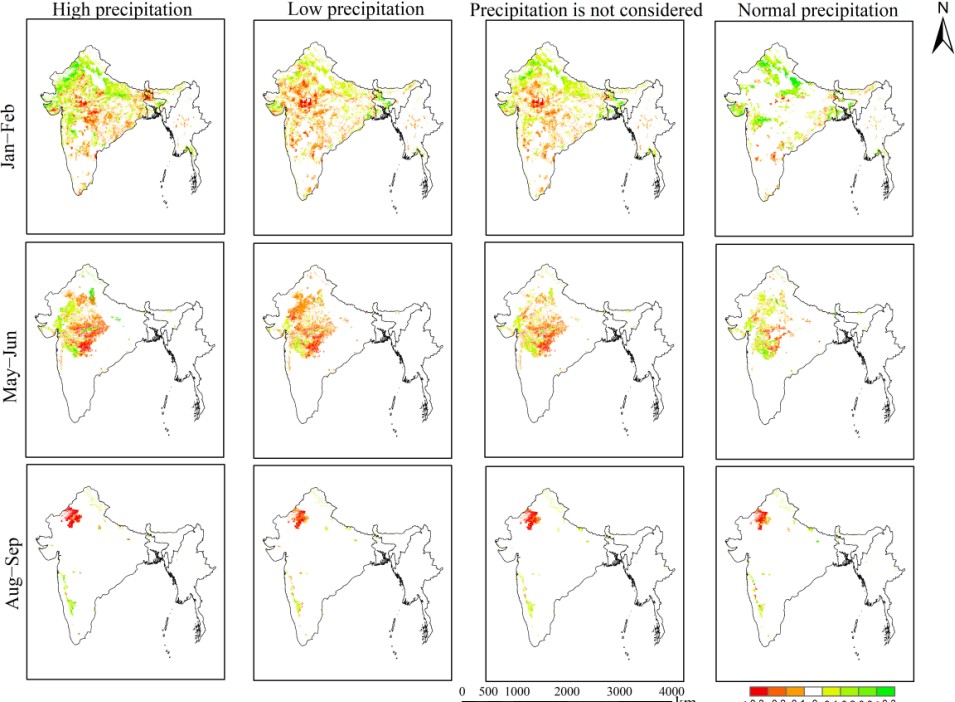

**Figure 8.** The degree of deviation of EVI between normal temperature with normal precipitation and different precipitation under extreme high temperature during growing seasons.

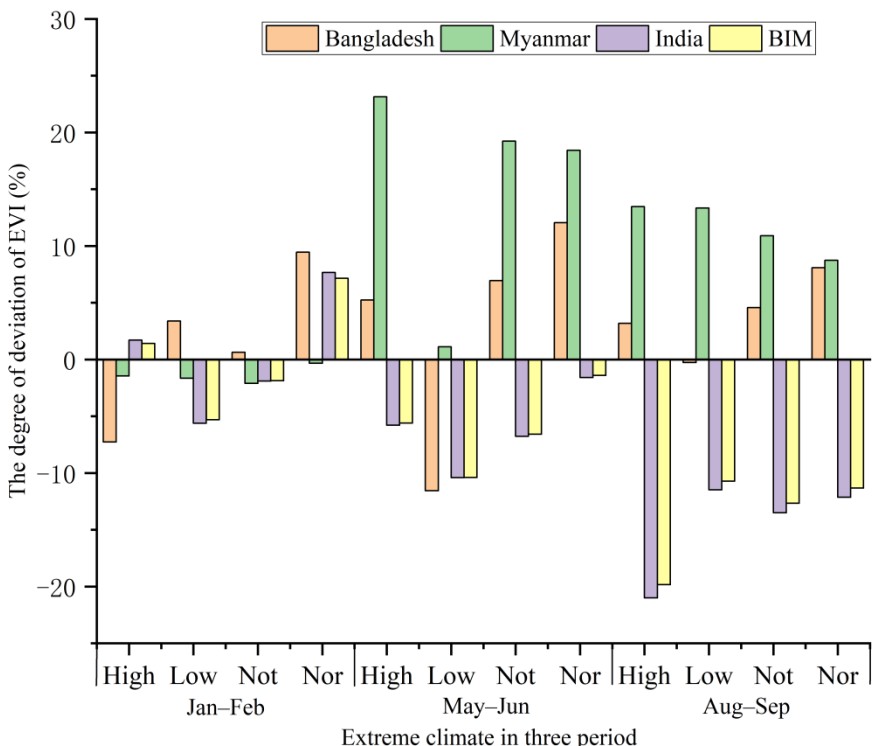

**Figure 9.** The degree of deviation of EVI under extreme climate in Bangladesh, India, Myanmar and South Asia (Abbreviations: High–Higher temperature and higher precipitation, Low–Higher temperature and lower precipitation, Not–Higher temperature without considering precipitation, Nor–Higher temperature and normal precipitation).

In this study, both NDVI and EVI presented similar results for the responses of the crop growth to extreme climate events. Regardless of whether precipitation is high or low, high temperatures mostly result in lower crop growth over large areas, especially in India. Here, India plays a dominant role with the largest cropland area in this study region.

It is worth mentioning that crops also positively respond to extreme high temperature events, and these positive responses are typically overlaid in areas with irrigated farmland; negative responses (Figures 6 and 8) are typically distributed at higher elevations such as mountains [85–88]. Exceptionally, in irrigated areas such as the Gangetic Plain and the Indus Plain, the EVI showed a significant negative response (Figure 6c,e), likely due to intensive irrigation that intensified the heat and humidity stress in the Indo-Gangetic Plain in wet years, increasing heat stress indicators and posing a serious threat to crops [89].

### 4.4. Adapting Measures to Climate Change

A historical assessment of extreme climate and its effects on crop yield are still needed to mitigate and adapt to climate change in the agriculture sector [90]. Seasonal adjustments to planting dates and crop varieties may be effective ways to combat increasing frequency and extent of climate changes [91–95]. Breeding new crop varieties through genetic modification to generate or accumulate genetic components that improve the adaptability of crops to one or more climatic stresses is an important method for long-term adaptation to stress conditions [96]. Furthermore, water management has been found to be relatively affordable and easier to implement on irrigated cropland [97]. Farmers are also becoming aware of the need to apply mixed cropping systems to minimize the risk of crop failure [98].

This study had some limitations. For example, the actual yield loss was not assessed due to the unavailability of on-the-ground yield measures. However, this study provides a fundamental understanding of impacts of extreme climate on crop growth through remote sensing-based measurements. Further crop model-based analysis and prediction are

essential for assessing the complex effects from extreme events interacting with multiple environmental factors [98,99], which is expected to help mitigate and adapt to climate change in the future for this area of Bangladesh, India and Myanmar, a prominent agricultural region of the world [100–106].

## 5. Conclusions

The frequency of climate extremes was found to be increasing and multiple events coincidently occur. Therefore, it is essential to quantify and assess the probability and extent of extreme events, and their effects on crop growth, to understand the full implications. Our results revealed that while the average probability of extreme temperatures is small, most regions (more than half) have experienced extreme events. At the same time, we applied remote sensing methods to provide a spatially explicit statistical assessment of the combined effects of extreme temperature under different precipitation levels on crops in South Asia. Although the probability of occurrence of dry years is greater than that of wet years, the harm caused by high temperature in wet years is far greater than that in dry years. From a geographical point of view, the areas with positive responses for crops under high temperature are mostly concentrated in irrigated cropland and negative responses are mostly in the mountains and other high-altitude regions. The event coincidence rate for extreme temperature and vegetation indices occurrence was investigated through event coincidence analysis. The three growing seasons displayed the highest densities of significant event coincidence rates at a low EVI for both high- and low-temperature extremes. In the future, further crop model research is essential to diagnose the mechanism of extreme climate impact on crop yields and predict possible crop yield loss under extreme climate change. This study provides some foundation for determining how extreme climate impacts crop growth, which is very important to be able to suggest actions for sustainable agricultural development while mitigating and adapting to climate change in the future.

**Supplementary Materials:** The following supporting information can be downloaded at: https://www.mdpi.com/article/10.3390/rs14236093/s1, Figure S1: Spatial patterns of significant event coincidence rates (SCRs) between different combinations of low and high temperature and Vegetation Index NDVI extremes during the three considered time periods(a, c, e and b, d, f represent extreme NDVI under extreme high and extreme low temperatures in Jan-Feb, May-Jun and Aug-Sep, respectively, with red representing extreme low NDVI values and green representing extreme high EVI values). Figure S2: The degree of deviation of NVI between normal temperature with normal precipitation and different precipitation under extreme high temperature during growing seasons. Figure S3: The degree of deviation of NDVI under extreme climate in Bangladesh, India, Myanmar and South Asia (Abbreviations: High–Higher temperature and higher precipitation, Low–Higher temperature and lower precipitation, Not–Higher temperature without considering precipitation, Nor–Higher temperature and normal precipitation).

**Author Contributions:** Writing—original draft, X.F.; Data curation, D.Z.; Formal analysis, X.S.; Methodology, J.W.; Supervision, M.W.; Investigation, S.W.; Writing—review & editing, A.E.W. All authors have read and agreed to the published version of the manuscript.

**Funding:** This research was funded by the National Natural Science Foundation of China (grant number: 31861143015, 42071373), and the Natural Science Foundation of Shandong Province, China (grant number: ZR2020MD021).

**Data Availability Statement:** The data used in this research will be available (by the corresponding author), upon reasonable request.

**Acknowledgments:** We thank the editor and three reviewers for their thoughtful comments that improved this research. We thank NASA, ERA and ESA for providing GIMMS and MODIS imagery, ERA5, MCD12Q1.

**Conflicts of Interest:** The authors declare no conflict of interest.

## Appendix A

Event consistency analysis is used to calculate the consistency between different types of events. In this paper, it is hypothesized that events in extreme Vegetation Index (*B*) precede events in extreme temperature (*A*), which are validated and tested.

Defining the consistency of the occurrence of two events first needs to determine the consistency in time. *A* pair of event time series *A* and *B* is here defined as two ordered event sets with timings $\left\{ t_1^A, \ldots, t_{i_A}^A \right\}$ and $\left\{ t_1^B, \ldots, t_{J_B}^B \right\}$ with number of events $i_A$, $J_B$, respectively. Both event series are assumed to cover a time interval $\left( t_0, t_f \right)$ of length $T = t_f - t_0$, such that $t_0 \leq t_1^A \leq \ldots \leq t_{i_A}^A \leq t_f$ and $t_0 \leq t_1^B \leq \ldots \leq t_{J_B}^B \leq t_f$.

When the symmetrical coincidence interval of *A* event is satisfied, and the assumption that *B* event must precede *A* event is relaxed, if

$$\left\lceil t_i^A - t_j^B \right\rceil \leq \Delta T \tag{A1}$$

Then the event coincidence analysis is satisfied.

Standard and composite measurements can be made for the consistency of occurrence between several events in a series of events available on a spatial grid or in different regions. This paper quantifies the aggregation consistency across all countries in the dataset, obtaining an overall measure of the strength of the relationship between the four types considered and its statistics, taking into account different assumptions. Similar to the case of single-pair event series, in a given set G of type *A* and type *B* events, the event coincidence rate is consistent for both cases. The aggregated precursor coincidence rate:

$$r_p^G(\Delta T, \tau) = \frac{\sum_{k \in G} \sum_{i=1}^{N_{A,k}} \Theta \left[ \sum_{j=1}^{N_{B,k}} 1_{[0, \Delta T]} \left( \left( t_i^{A,K} - \tau \right) - t_j^{B,K} \right) \right]}{\sum_{k \in G} N_{A,k}} \tag{A2}$$

These events are normalized by the maximum possible number of such events to measure the total number of all events or events occurring in all paired sequences of events. In the same situation, the aggregated trigger coincidence rate:

$$r_t^G(\Delta T, \tau) = \frac{\sum_{k \in G} \sum_{i=1}^{N_{B,k}} \Theta \left[ \sum_{j=1}^{N_{A,k}} 1_{[0, \Delta T]} \left( \left( t_i^{A,K} - \tau \right) - t_j^{B,K} \right) \right]}{\sum_{k \in G} N_{B,k}} \tag{A3}$$

This is consistent with the normalized total number of trigger coincidences that occurred across all pairwise events. The above algorithm was accomplished through Python-based code and the code could be shared with the requisition.

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
