# Peer review of "Impacts of Extreme Temperature and Precipitation on Crops during the Growing Season in South Asia"

_remotesensing, doi:10.3390/rs14236093_

Round 1

Reviewer 1 Report (Previous Reviewer 1)

I notice  that all my last comments have been revised. Current manuscript can be considered for accepting after correcting following questions: 

1. In the abstract, the likelihood of extreme precipitation occurring on crops does not seem to be analyzed with event coincidence analysis.

2. In line 127, the VI data is from GIMMS, does it refer to NDVI only?

3. Lines 350-351, the statement is messed up and needs to be readjusted.

Author Response

General comments

I notice that all my last comments have been revised. Current manuscript can be considered for accepting after correcting following questions.

  • Reply: Thank you very much for your recognition of the value of this study.

Reviewer 1.1. In the abstract, the likelihood of extreme precipitation occurring on crops does not seem to be analyzed with event coincidence analysis.

  • Reply: Thank you very much for this suggestion. Yes, in this study, event coincidence analysis was done only for extreme temperature, which was clarified in the abstract, now in lines 18-20.

Reviewer 1.2. In line 127, the VI data is from GIMMS, does it refer to NDVI only?

  • Reply: The VI data includes both GIMMS-based NDVI and MODIS-based EVI and both data were analyzed in this study and their respective advantages considered.. However, the results of EVI were mainly presented and discussed, as suggested strongly by another reviewer, while the results from NDVI were shown in supplemental Figures 1 and 2. Another reviewer felt this would allow us to present the results more clearly and concisely. For details, please see lines 66-79 and 123-133 in this version.

Reviewer 1.3. Lines 350-351, the statement is messed up and needs to be readjusted.

  • Reply: Thank you for such a detailed suggestion, we have polished the sentence and made it more clear. Please see details in lines 351-352.

Reviewer 2 Report (Previous Reviewer 2)

I have read the revised manuscript that mainly presented the application of low orbit satellite based imaging to identify impacts of extreme temperature and precipitation on crops during the growing season in South Asia. Thanks for incorporating all previous comments and clarifying doubts, specially related to vegetation indices used. I accept this manuscript with the MDPI publishing standards.

Author Response

Thank you very much for your recognition of our efforts to respond to all previous review comments.

Reviewer 3 Report (Previous Reviewer 3)

The authors of submission remotesensing-2001619 present an interesting work on impact of extreme weather on crops during the growing season in South Asia, mainly the Indian peninsula and the Bay of Bengal. Their work is based on public data from institutions such as Global Inventory Monitoring and Modeling System. The datasets included data collected within different time, ranging between 1979-2020. The data comes from different sensors, has different spatial resolutions and relates to different characteristics, such as temperature, precipitation, or land use. It seems that the paper is a resubmission, but it is completely unreadable in the presented state. The paper as a new submission should not include the same exact content that was rejected in the previous approach. It needs to be clarified. I would recommend a major revision. 

Author Response

Thank you very much for your careful review, we have made careful revisions. Yes, this work is based on public data including MODIS EVI, GIMMS-NDVI, ERA5 climate data, etc. and the remote sensing came from different sensors with different spatial resolutions to diagnose the impacts of extreme temperature events on crop growth. It would have been nice to have all attributes measured exactly the same over the same time period, but we feel this does not distract from the value of these data to provide insight into a complicated climate issue. Maybe this will lead to more consistent measures into the future. However, both EVI and NDVI provide a unique chance to analyze extreme temperature and precipitation effects on crop growth quantified by the Vegetation Index with their respective strengths. For this paper, the results are primarily determined from examining EVI-based analyses. The GIMMS-based NDVI were also analyzed, separately, considering those data have a long-term observation spanning 1981 to 2015 and improved data quality by accounting for biases such as calibration loss, orbital drift, volcanic eruptions (Pinzon and Tucker 2014, Tucker et al. 2005), etc. In order to present the results more clearly and concisely, the results from NDVI is shown in supplemental Figures 1 and 2. Therefore, we clarified that both EVI and NDVI data-based results were presented respectively in the main text and the supplemental figures in this study.

Round 2

Reviewer 3 Report (Previous Reviewer 3)

The readability of the article has been significantly improved. Thank you for addressing all my comments and questions. I don't see any major errors, the paper is ready for publication.

This manuscript is a resubmission of an earlier submission. The following is a list of the peer review reports and author responses from that submission.

Round 1

Reviewer 1 Report

This study found complex correlation patterns between extreme heat / precipitation and the influences on rice and wheat in South Asian. It has great implications on guiding farmers in response to climate changes. 

(1) Title: the extreme weather included extreme temperature and precipitation, and it is suggested that the authors should consider revising the title appropriately, such as, extreme temperature and precipitation?

(2) The authors should replace the Fig.1 with higher resolution.

(3) Abbreviations appearing in the figures/tables should be described with more details in captures of all the figures/tables to enhance self-explanatoriness. For example, it is difficult to understand the abbreviations in Table 1 and Table 3; Figure 2 shows that the proportion of extreme weather events in different months and regions is also difficult to understand for readers. The authors may consider other type figures to describe relevant results.

(4) The last paragraph of results 3.1 can be considered to remove to the section of discussion.

(5) Figure 3a, 3b existed but I can't find figure3c, d, f, g in the figure 3. 

(6) Some figures and paragraphs in Discussion 4.2 and 4.3 should be placed in appropriate position in results.

(7) The authors should delete or updated the irrelevant words in the end of the manuscript and conform to the standard manuscript submission format. Such as: “Supplementary Materials”, “Author Contributions”.

Author Response

Reviewer: 1

Comments to the Author

Comments to remotesensing-1896748

General comments

This study found complex correlation patterns between extreme heat / precipitation and the influences on rice and wheat in South Asian. It has great implications on guiding farmers in response to climate changes.

  • Reply: Thank you very much for your careful review, our team appreciated all the specific comments and we addressed all of these specific suggestions. Especially, our co-author, Alan E. Watson, has conducted multiple rounds of review on this manuscript and very carefully polished the grammar of this manuscript before this submission.

Reviewer 1.1. Title: the extreme weather included extreme temperature and precipitation, and it is suggested that the authors should consider revising the title appropriately, such as, extreme temperature and precipitation?

  • Reply: Thank you very much for your suggestion. We have revised the title to “Impacts of extreme temperature and precipitation on crops during the growing season in South Asia”.

Reviewer 1.2. The authors should replace the Fig.1 with higher resolution.

  • Reply: Thank you for such a detailed suggestion, we have replaced it with a higher quality figure. Please see Figure 1.

Reviewer 1.3. Abbreviations appearing in the figures/tables should be described with more details in captures of all the figures/tables to enhance self-explanatoriness. For example, it is difficult to understand the abbreviations in Table 1 and Table 3; Figure 2 shows that the proportion of extreme weather events in different months and regions is also difficult to understand for readers. The authors may consider other type figures to describe relevant results.

  • Reply: Thank you for such a detailed suggestion, we have explained and checked all abbreviations in the whole MS, including those in Table 1 in lines 107-115 on page 3 (section 2.1) of the latest revision and the abbreviations in Table 3 are explained in lines 167-175 on page 6 (section 2.3.1). Further explanation of Original Picture 2 (now Picture 3) was added on pages 8-9, lines 254-267.

Reviewer 1.4. The last paragraph of results 3.1 can be considered to remove to the section of discussion.

  • Reply: Thank you very much for your suggestion. We have revised it around lines 404-406 of page 14.

Reviewer 1.5. Figure 3a, 3b existed but I can't find figure3c, d, f, g in the figure 3.

  • Reply: Thank you very much for your question. I'm sorry for my mistake, we have revised it.

Reviewer 1.6. Some figures and paragraphs in Discussion 4.2 and 4.3 should be placed in appropriate position in results.

  • Reply: Thank you very much for your suggestion. We adjusted Figure 6 (the new revised version is Figure 5) into Results 3.2.

Reviewer 1.7. The authors should delete or updated the irrelevant words in the end of the manuscript and conform to the standard manuscript submission format. Such as: “Supplementary Materials”, “Author Contributions”.

  • Reply: Thank you very much for your suggestion. We have removed irrelevant text at the end of the new manuscript.

Reviewer 2 Report

In my opinion, the overall paper is well written to read and understand, except abstract. The topic seems actual, since high quality of data is required for further processing steps, such as classification or variable retrievals. The results provided in this paper are quite interesting, and the paper is technically correct. The paper seems well written in an understandable way. The English language of the manuscript should be improved. Some place there are too confusing symbols which can be clarify more precisely. In my opinion, the paper can only publish after addressing following minor comments,

1.      The structure of the manuscript is bit confusing. I suggest the authors to add a schematic view of the used methodology in order to clarify the content.

2.      I suggest the authors to explain more about the image pre-processing. Did you apply atmospheric correction? What are the pre-processing that you have applied to the images?

3.      All abbreviations should be properly explain on it’s first use, e.g. what is NDVI3g, ERA5, MCD12Q1 in section 2.2?

4.      In line 155 to 165, I think that there is a problem with the notation. It seems that the value of parameter P was chosen arbitrarily, Please elaborate more on your decision to define P.

5.      Why NDVI and EVI vegetation index were only selected to evaluate the distinct responses of major crops to climate extremes? 

6.      How you defined the values for event coincidence analysis (ECA) in the proposed model?

7.      On what basis you minimized the mean absolute percentage error of prediction for yield estimation accuracy caused by extreme climate?

8.      How you define the ECA and VI validation to simplify the model framework?

9.      What is the observation characteristic of combining NDVI and EVI indices? Please define properly in the text.

10.  Define the formulation of models used in Table 5.

11.   Present an explicit and clear algorithmic steps used in this study data simulation.

12.  Use the high resolution image for Figure-3 and Figure-4.

13.  There is a need to reduce significant unrelated references and to add couple of proper references to support this study.

Also, the paper should be proofread for sentences flow, English grammar correction, and spelling mistakes.

Author Response

Reviewer: 2

Comments to the Author

In my opinion, the overall paper is well written to read and understand, except abstract. The topic seems actual, since high quality of data is required for further processing steps, such as classification or variable retrievals. The results provided in this paper are quite interesting, and the paper is technically correct. The paper seems well written in an understandable way. The English language of the manuscript should be improved. Some place there are too confusing symbols which can be clarify more precisely. In my opinion, the paper can only publish after addressing following minor comments,

  • Reply: Thank you very much for your careful review, our team appreciated all the specific comments and we addressed all of these specific suggestions. Especially, our co-author, Alan E. Watson, has conducted multiple rounds of review on this manuscript and very carefully polished the grammar of this manuscript before this submission.

Reviewer 1.1.  The structure of the manuscript is bit confusing. I suggest the authors to add a schematic view of the used methodology in order to clarify the content.

  • Reply: Based on your suggestion, a framework on the method was prepared and provided as Figure2.

Reviewer 1.2. I suggest the authors to explain more about the image pre-processing. Did you apply atmospheric correction? What are the pre-processing that you have applied to the images?

  • Reply: Thank you very much for your suggestion. The remote sensing products used are GIMMS NDVI3g, MOD13C2, and ESA CCI. The GIMMS NDVI3g data set was processed in a way consistent with and quantitatively comparable to NDVI derived from improved sensors such as MODIS and SPOT-4 Vegetation, and was corrected for dropped scan lines, navigation errors, data drop outs, edge-of-orbit composite discontinuities, and other artifacts (Tucker et al. 2005); MOD13C2 products have been corrected for atmospheric gases, aerosols, thin cirrus clouds, water vapor and ozone (Huete et al. 2002) ; ESA CCI has also completed preprocessing such as atmospheric calibration and geometric correction (Bontemps et al. 2013) . Therefore, the data were directly applied in this study after we performed projection transformation and image clipping. The pre-processing steps have been described and can now be found in lines 133-152 of page 5 in section 2.2.

Reviewer 1.3. All abbreviations should be properly explain on it’s first use, e.g. what is NDVI3g, ERA5, MCD12Q1 in section 2.2?

  • Reply: Thank you very much, we have revised it around lines 133-152 of page 5

Reviewer 1.4. In line 155 to 165, I think that there is a problem with the notation. It seems that the value of parameter P was chosen arbitrarily, Please elaborate more on your decision to define P.

  • Reply: Thank you very much for your question, we define the percentage of monthly precipitation anomalies as P, explained in lines 156-175 pages 5-6 in section 2.3.1. And checked all of such problems in the whole manuscription.

Reviewer 1.5. Why NDVI and EVI vegetation index were only selected to evaluate the distinct responses of major crops to climate extremes?

  • Reply: Thank you very much for your question. On the one hand, the time series of GIMMS NDVI3g is the longest dataset extending from 1980 to 2015, which can more accurately reflect the impact of extreme climate on crops (Wang et al. 2014; Ye et al. 2021). On the other hand, the EVI in the MODIS product is more sensitive to extreme climate than the NDVI from the same sensor (Son et al. 2014). Based on above, those two datasets were applied in this study, with improved justification found around Lines 70-77 of Page 2.

Reviewer 1.6. How you defined the values for event coincidence analysis (ECA) in the proposed model?

  • Reply: Thank you very much for your question. According to (Donges et al. 2016; Odenweller 2020; Sulla-Menashe and Friedl 2018), first the extreme temperatures (vegetation index) was defined as the 10th and 90th quantiles. Then the ECA package in Python was applied to perform the ECA. Those were described around lines 185 - 203 of page 6.

Reviewer 1.7. On what basis you minimized the mean absolute percentage error of prediction for yield estimation accuracy caused by extreme climate?

  • Reply: Thank you very much for your question. However, our research was not based on yield prediction, and the impact of extreme temperature and precipitation on crop were quantified through NDVI and EVI. The impact is mainly measured through the comparison VI at the normal situation with the corresponding value at the different extreme conditions in the same time period, which could be found in Figure 4 and Figure 5.

Reviewer 1.8. How you define the ECA and VI validation to simplify the model framework?

  • Reply: Thank you very much for your question. The method based on the ECA and VI have been applied in the previous study (Baumbach et al. 2017). Here we directly applied this well-established method in this study, which could be found around Lines 209-214 of Pages 6-7.

Reviewer 1.9. What is the observation characteristic of combining NDVI and EVI indices? Please define properly in the text.

  • Reply: Thank you very much for your question. We mainly study the characteristics of NDVI and EVI in extreme climates. First of all, different indices themselves will have certain differences and lack comparability, but in the same time period, the same index at the different situations have good comparability. From Figures 3 and 4, we can see that, compared with normal climate conditions, high temperatures accompanied by wet years can cause the lower VI values than that at normal condition.

Reviewer 1.10. Define the formulation of models used in Table 5.

  • Reply: Thank you very much for your suggestion. The formulations on the event coincidence analysis (ECA) were well described in a previous study(Baumbach et al. 2017) and appended to this MS by considering its complex processes and lots of equations. For the specific formulas, please refer to Appendix A

Reviewer 1.11. Present an explicit and clear algorithmic steps used in this study data simulation.

  • Reply: Thank you very much for your suggestion. The Event Coincidence Analysis (ECA) in this article was accomplished through using some packages in Python and we can share the codes if it would be requested. However, the algorithm and some key equations were appended to the end the MS.

Reviewer 1.12. Use the high resolution image for Figure-3 and Figure-4.

  • Reply: Thank you for the suggestion, we have replaced those with the higher quality figures (Figure-3 and Figure-4).

Reviewer 1.13. There is a need to reduce significant unrelated references and to add couple of proper references to support this study.

  • Reply: Thank you for the suggestion. We checked the reference and ensure to cite all important literatures on this research field in this study.

Reviewer 3 Report

The authors of submission remotesensing-1896748 present an interesting work on impact of extreme weather on crops during the growing season in South Asia, mainly the Indian peninsula and the Bay of Bengal. Their work is based on public data from institutions such as Global Inventory Monitoring and Modeling System. The datasets included data collected within different time, ranging between 1979-2020. The data comes from different sensors, has different spatial resolutions and relates to different characteristics, such as temperature, precipitation, or land use. They performed a literature review, which is up to date, but in my opinion should be extended and focused more on the technical side. The method and results of the analysis are described in detail, but it lacks a technical description of the analysis. The great example is subchapter 3.1 Probabilities of extreme climate events, what is the probabilistic model that you used? How you calculated and analyzed these findings? The topic is of interest in that it has the potential to be used as a case study. However the authors completely abandoned description of the technical side of their analysis. Overall, the paper is well-structured, readable, but non-coherent and lacks technical descriptions. It need to be clarified. I would recommend a major revision.

General comments:

§  The quality of the data has not been discussed;

§  The adopted technical approach and engineering process behind their models were not discussed at all;

§  The process of implementation should be described more precisely;

§  You should improve the “Conclusions” part. It is too short for such an extensive study;

§  You need to clarify what exactly is your unique contribution;

§  The paper contains some stylistic errors. You should try to avoid wordiness and limit the use of passive voice to improve readability. I recommend proofreading.

Additional questions:

·       Did you perform some sensitivity tests to evaluate the adopted probabilistic model?

Author Response

Reviewer: 3

Comments to the Author

The authors of submission remotesensing-1896748 present an interesting work on impact of extreme weather on crops during the growing season in South Asia, mainly the Indian peninsula and the Bay of Bengal. Their work is based on public data from institutions such as Global Inventory Monitoring and Modeling System. The datasets included data collected within different time, ranging between 1979-2020. The data comes from different sensors, has different spatial resolutions and relates to different characteristics, such as temperature, precipitation, or land use. They performed a literature review, which is up to date, but in my opinion should be extended and focused more on the technical side. The method and results of the analysis are described in detail, but it lacks a technical description of the analysis. The great example is subchapter 3.1 Probabilities of extreme climate events, what is the probabilistic model that you used? How you calculated and analyzed these findings? The topic is of interest in that it has the potential to be used as a case study. However the authors completely abandoned description of the technical side of their analysis. Overall, the paper is well-structured, readable, but non-coherent and lacks technical descriptions. It need to be clarified. I would recommend a major revision.

  • Reply: Thank you very much for your careful review, our team appreciated all the specific comments and we addressed all of these specific suggestions. Especially, our co-author, Alan E. Watson, has conducted multiple rounds of review on this manuscript and very carefully polished the grammar of this manuscript before this submission.

Reviewer 1.1. The quality of the data has not been discussed;

  • Reply: Thank you for the suggestion. The data used in the study have been processed and published as data products. In this study, we explained the quality of those data in lines 133-152 of page 5 in section 2.2. Therefore, the data were directly applied in this study after we performed projection transformation and image clipping.

Reviewer 1.2. The adopted technical approach and engineering process behind their models were not discussed at all;

  • Reply: Thank you for the suggestion. The methods in this MS was accomplished through using some packages in Python. It contained a large number of complex equations and the specific processes and formulas are shown in Appendix A.

Reviewer 1.3. The process of implementation should be described more precisely;

  • Reply: Thank you for the suggestion. We have made further illustration on the method and data, which can be found in section 2.3.1. And a framework on the method was provided as Figure 2.

Reviewer 1.4. You should improve the “Conclusions” part. It is too short for such an extensive study;

  • Reply: Thank you for the suggestion, we have revised it around lines 474-485 of pages 17-18.

Reviewer 1.5. You need to clarify what exactly is your unique contribution;

  • Reply: Thank you for the suggestion. One of contribution lies in the use of a joint probabilistic approach to investigate the combined effects of extreme temperatures and precipitation on crop yields, and it was discussed around lines 78-84 of page 2.

Reviewer 1.6. The paper contains some stylistic errors. You should try to avoid wordiness and limit the use of passive voice to improve readability. I recommend proofreading

  • Reply: Thank you for the suggestion. Our co-author, Alan E. Watson, has conducted multiple rounds of review on this manuscript and very carefully polished the grammar of this manuscript before the submission.

Additional questions: Did you perform some sensitivity tests to evaluate the adopted probabilistic model?

  • Reply: Thank you for the suggestion. We used the Analysis of variance (ANOVA) to test the statistic significant of the impacts of different extreme temperature condition on the crop (Table 4) and indicated that extreme temperature events significantly impact crop growth as quantified by vegetation index. The analysis indirectly illustrated the adopted probabilistic model is applicable, which was discussed around lines 272-282.
